# SUMO: Unbiased Estimation of Log Marginal Probability for Latent Variable Models

**Yucen Luo**[*]
Tsinghua University
luoyc15@mails.tsinghua.edu.cn

**Alex Beatson**
Princeton University
abeatson@cs.princeton.edu

**Mohammad Norouzi**
Google Research
mnorouzi@google.com

**Jun Zhu**
Tsinghua University
dcszj@tsinghua.edu.cn

**David Duvenaud**
University of Toronto
duvenaud@cs.toronto.edu

**Ryan P. Adams**
Princeton University
rpa@princeton.edu

**Ricky T. Q. Chen**[*]
University of Toronto
rtqichen@cs.toronto.edu

## Abstract

Standard variational lower bounds used to train latent variable models produce biased estimates of most quantities of interest. We introduce an unbiased estimator of the log marginal likelihood and its gradients for latent variable models based on randomized truncation of infinite series. If parameterized by an encoder-decoder architecture, the parameters of the encoder can be optimized to minimize its variance of this estimator. We show that models trained using our estimator give better test-set likelihoods than a standard importance-sampling based approach for the same average computational cost. This estimator also allows use of latent variable models for tasks where unbiased estimators, rather than marginal likelihood lower bounds, are preferred, such as minimizing reverse KL divergences and estimating score functions.

## 1 Introduction

Latent variable models are powerful tools for constructing highly expressive data distributions and for understanding how high-dimensional observations might possess a simpler representation. Latent variable models are often framed as probabilistic graphical models, allowing these relationships to be expressed in terms of conditional independence. Mixture models, probabilistic principal component analysis (Tipping & Bishop, 1999), hidden Markov models, and latent Dirichlet allocation (Blei et al., 2003) are all examples of powerful latent variable models. More recently there has been a surge of interest in probabilistic latent variable models that incorporate flexible nonlinear likelihoods via deep neural networks (Kingma & Welling, 2014). These models can blend the advantages of highly structured probabilistic priors with the empirical successes of deep learning (Johnson et al., 2016; Luo et al., 2018). Moreover, these explicit latent variable models can often yield relatively interpretable representations, in which simple interpolation in the latent space can lead to semantically-meaningful changes in high-dimensional observations (e.g., Higgins et al. (2017)).

It can be challenging, however, to fit the parameters of a flexible latent variable model, since computing the marginal likelihood of the data requires integrating out the latent variables. Typical approaches to this problem include the celebrated expectation maximization algorithm (Dempster et al., 1977), Markov chain Monte Carlo, and the Laplace approximation. Variational inference generalizes expectation maximization by forming a lower bound on the aforementioned (log) marginal likelihood, using a tractable approximation to the unmanageable posterior over latent variables. The maximization of this lower bound—rather than the true log marginal likelihood—is often relatively straightforward when using automatic differentiation and Monte Carlo sampling. However, a

---

[*]Equal contribution.

lower bound may be ill-suited for tasks such as posterior inference and other situations where there exists an entropy maximization objective; for example in entropy-regularized reinforcement learning (Williams & Peng, 1991; Mnih et al., 2016; Norouzi et al., 2016) which requires minimizing the log probability of the samples under the model.

While there is a long history in Bayesian statistics of estimating the marginal likelihood (e.g., Newton & Raftery (1994); Neal (2001)), we often want high-quality estimates of the *logarithm* of the marginal likelihood, which is better behaved when the data is high dimensional; it is not as susceptible to underflow and it has gradients that are numerically sensible. However, the log transformation introduces some challenges: Monte Carlo estimation techniques such as importance sampling do not straightforwardly give unbiased estimates of this quantity. Nevertheless, there has been significant work to construct estimators of the log marginal likelihood in which it is possible to explicitly trade off between bias against computational cost (Burda et al., 2016; Bamler et al., 2017; Nowozin, 2018). Unfortunately, while there are asymptotic regimes where the bias of these estimators approaches zero, it is always possible to optimize the parameters to increase this bias to infinity.

In this work, we construct an unbiased estimator of the log marginal likelihood. Although there is no theoretical guarantee that this estimator has finite variance, we find that it can work well in practice. We show that this unbiased estimator can train latent variable models to achieve higher test log-likelihood than lower bound estimators at the same expected compute cost. More importantly, this unbiased estimator allows us to apply latent variable models in situations where these models were previously problematic to optimize with lower bound estimators. Such applications include latent variable modeling for posterior inference and for reinforcement learning in high-dimensional action spaces, where an ideal model is one that is highly expressive yet efficient to sample from.

## 2 PRELIMINARIES

### 2.1 LATENT VARIABLE MODELS

Latent variable models (LVMs) describe a distribution over data in terms of a mixture over unobserved quantities. Let $p_\theta(x)$ be a family of probability density (mass) functions on a data space $\mathcal{X}$, indexed by parameters $\theta$. We will generally refer to this as a "density" for consistency, even when the data should be understood to be discrete; similarly we will use integrals even when the marginalization is over a discrete set. In a latent variable model, $p_\theta(x)$ is defined via a space of latent variables $\mathcal{Z}$, a family of mixing measures on this latent space with density denoted $p_\theta(z)$, and a conditional distribution $p_\theta(x \,|\, z)$. This conditional distribution is sometimes called an "observation model" or a conditional likelihood. We will take $\theta$ to parameterize both $p_\theta(x \,|\, z)$ and $p_\theta(z)$ in the service of determining the marginal $p_\theta(x)$ via the mixture integral:

$$p_\theta(x) := \int_{\mathcal{Z}} p_\theta(x \,|\, z) p_\theta(z) \, dz = \mathbb{E}_{z \sim p_\theta(z)} \left[ p_\theta(x \,|\, z) \right] . \tag{1}$$

This simple formalism allows for a large range of modeling approaches, in which complexity can be baked into the latent variables (as in traditional graphical models), into the conditional likelihood (as in variational autoencoders), or into both (as in structured VAEs). The downside of this mixing approach is that the integral may be intractable to compute, making it difficult to evaluate $p_\theta(x)$— a quantity often referred to in Bayesian statistics and machine learning as the *marginal likelihood* or *evidence*. Various Monte Carlo techniques have been developed to provide consistent and often unbiased estimators of $p_\theta(x)$, but it is usually preferable to work with $\log p_\theta(x)$ and unbiased estimation of this quantity has, to our knowledge, not been previously studied.

### 2.2 TRAINING LATENT VARIABLE MODELS

Fitting a parametric distribution to observed data is often framed as the minimization of a difference between the model distribution and the empirical distribution. The most common difference measure is the forward Kullback-Leibler (KL) divergence; if $p_{\mathsf{data}}(x)$ is the empirical distribution and $p_\theta(x)$ is a parametric family, then minimizing the KL divergence ($D_{\mathrm{KL}}$) with respect to $\theta$ is equivalent to maximizing the likelihood:

$$D_{\mathrm{KL}}(p_{\mathsf{data}} \,\|\, p_\theta) = \int_{\mathcal{X}} p_{\mathsf{data}}(x) \log \frac{p_{\mathsf{data}}(x)}{p_\theta(x)} \, dx = -\mathbb{E}_{\mathsf{data}} \left[ \log p_\theta(x) \right] + \mathrm{const} . \tag{2}$$

Equivalently, the optimization problem of finding the MLE parameters $\theta$ comes down to maximizing the expected log probability of the data:

$$\theta^{\mathsf{MLE}} = \arg\min_{\theta} D_{\mathrm{KL}}(p_{\mathsf{data}} \,\|\, p_\theta) = \arg\max_{\theta} \mathbb{E}_{\mathsf{data}}\left[\log p_\theta(x)\right]. \tag{3}$$

Since expectations can be estimated in an unbiased manner using Monte Carlo procedures, simple subsampling of the data enables powerful stochastic optimization techniques, with stochastic gradient descent in particular forming the basis for learning the parameters of many nonlinear models. However, this requires unbiased estimates of $\nabla_\theta \log p_\theta(x)$, which are not available for latent variable models. Instead, a stochastic lower bound of $\log p_\theta(x)$ is often used and then differentiated for optimization.

Though many lower bound estimators (Burda et al., 2016; Bamler et al., 2017; Nowozin, 2018) are applicable, we focus on an importance-weighted evidence lower bound (Burda et al., 2016). This lower bound is constructed by introducing a proposal distribution $q(z; x)$ and using it to form an importance sampling estimate of the marginal likelihood:

$$p_\theta(x) = \int_{\mathcal{Z}} p_\theta(x \mid z)\, p_\theta(z)\, dz = \int_{\mathcal{Z}} q(z; x) \frac{p_\theta(x \mid z)\, p_\theta(z)}{q(z; x)}\, dz = \mathbb{E}_{z \sim q}\left[\frac{p_\theta(x \mid z)\, p_\theta(z)}{q(z; x)}\right]. \tag{4}$$

If $K$ samples are drawn from $q(z; x)$ then this provides an unbiased estimate of $p_\theta(x)$ and the biased "importance-weighted autoencoder" estimator $\mathrm{IWAE}_K(x)$ of $\log p_\theta(x)$ is given by

$$\mathrm{IWAE}_K(x) := \log \frac{1}{K} \sum_{k=1}^{K} \frac{p_\theta(x \mid z_k)\, p_\theta(z_k)}{q(z_k; x)}, \quad z_k \stackrel{iid}{\sim} q(z; x). \tag{5}$$

The special case of $K = 1$ generates an unbiased estimate of the evidence lower bound (ELBO), which is often used for performing variational inference by stochastic gradient descent. While the IWAE lower bound acts as a useful replacement of $\log p_\theta(x)$ in maximum likelihood training, it may not be suitable for other objectives such as those that involve entropy maximization. We discuss tasks for which a lower bound estimator would be ill-suited in Section 3.4.

There are two properties of IWAE that will allow us to modify it to produce an unbiased estimator: First, it is consistent in the sense that as the number of samples $K$ increases, the expectation of $\mathrm{IWAE}_K(x)$ converges to $\log p_\theta(x)$. Second, it is also monotonically non-decreasing in expectation:

$$\log p_\theta(x) = \lim_{K \to \infty} \mathbb{E}[\mathrm{IWAE}_K(x)] \qquad \text{and} \qquad \mathbb{E}[\mathrm{IWAE}_{K+1}(x)] \geq \mathbb{E}[\mathrm{IWAE}_K(x)]. \tag{6}$$

These properties are sufficient to create an unbiased estimator using the Russian roulette estimator.

## 2.3 RUSSIAN ROULETTE ESTIMATOR

In order to create an unbiased estimator of the log probability function, we employ the Russian roulette estimator (Kahn, 1955). This estimator is used to estimate the sum of infinite series, where each sample of the estimator almost surely requires only a finite amount of computation. Intuitively, the Russian roulette estimator relies on a randomized truncation and upweighting of each term to account for the possibility of not computing the remaining terms.

To illustrate the idea, let $\tilde{\Delta}_k$ denote the $k$-th term of an infinite series. Assume the partial sum of the series $\sum_{k=1}^{\infty} \tilde{\Delta}_k$ converges to some quantity we wish to obtain. We can construct a simple estimator by always computing the first term then flipping a coin $b \sim \mathrm{Bernoulli}(q)$ to determine whether we stop or continue evaluating the remaining terms. With probability $1 - q$, we compute the rest of the series. By reweighting the remaining future terms by $1/(1-q)$, we obtain an unbiased estimator:

$$\tilde{Y} = \tilde{\Delta}_1 + \left(\frac{\sum_{k=2}^{\infty} \tilde{\Delta}_k}{1 - q}\right) \mathbb{1}_{b=0} + (0)\mathbb{1}_{b=1} \qquad \mathbb{E}[\tilde{Y}] = \tilde{\Delta}_1 + \frac{\sum_{k=2}^{\infty} \tilde{\Delta}_k}{1 - q}(1 - q) = \sum_{k=1}^{\infty} \tilde{\Delta}_k.$$

To obtain the "Russian roulette" (RR) estimator (Forsythe & Leibler, 1950), we repeatedly apply this trick to the remaining terms. In effect, we make the number of terms a random variable $\mathcal{K}$, taking values in $1, 2, \ldots$ to use in the summation (i.e., the number of coin flips) from some distribution

with probability mass function $p(K) = \mathbb{P}(\mathcal{K} = K)$ with support over the positive integers. With $K$ drawn from $p(K)$, the estimator takes the form:

$$\hat{Y}(K) = \sum_{k=1}^{K} \frac{\tilde{\Delta}_k}{\mathbb{P}(\mathcal{K} \geq k)} \qquad\qquad \mathbb{E}_{K \sim p(K)}[\hat{Y}(K)] = \sum_{k=1}^{\infty} \tilde{\Delta}_k\,. \qquad (7)$$

The equality on the right hand of equation 7 holds so long as (i) $\mathbb{P}(\mathcal{K} \geq k) > 0,\ \forall k > 0$, and (ii) the series converges absolutely, i.e., $\sum_{k=1}^{\infty} |\tilde{\Delta}_k| < \infty$ (Chen et al. (2019); Lemma 3). This condition ensures that the average of multiple samples will converge to the value of the infinite series by the law of large numbers. However, the variance of this estimator depends on the choice of $p(K)$ and can potentially be very large or even infinite (McLeish, 2011; Rhee & Glynn, 2015; Beatson & Adams, 2019).

# 3 SUMO: UNBIASED ESTIMATION OF LOG PROBABILITY FOR LVMS

## 3.1 RUSSIAN ROULETTE TO DEBIAS LOWER BOUNDS

We can turn any absolutely convergent series into a telescoping series and apply the Russian roulette randomization to form an unbiased stochastic estimator. We focus here on the IWAE bound described in Section 2.2. Let $\Delta_k(x) = \text{IWAE}_{k+1}(x) - \text{IWAE}_k(x)$, then since $\mathbb{E}_q[\Delta_k(x)]$ converges absolutely, we apply equation 7 to construct our estimator, which we call SUMO (Stochastically Unbiased Marginalization Objective). The detailed derivation of SUMO is in Appendix A.1.

$$\text{SUMO}(x) = \text{IWAE}_1(x) + \sum_{k=1}^{K} \frac{\Delta_k(x)}{\mathbb{P}(\mathcal{K} \geq k)} \qquad \text{where} \quad K \sim p(K)\,. \qquad (8)$$

The randomized truncation of the series using the Russian roulette estimator means that this is an unbiased estimator of the log marginal likelihood, regardless of the distribution $p(K)$:

$$\mathbb{E}\left[\text{SUMO}(x)\right] = \log p_\theta(x)\,, \qquad (9)$$

where the expectation is taken over $p(K)$ and $q(z; x)$ (see Algorithm 1 for our exact sampling procedure). Furthermore, under some conditions, we have $\mathbb{E}\left[\nabla_\theta \text{SUMO}(x)\right] = \nabla_\theta \mathbb{E}\left[\text{SUMO}(x)\right] = \nabla_\theta \log p_\theta(x)$ (see Appendix A.4).

## 3.2 OPTIMIZING VARIANCE-COMPUTE PRODUCT BY CHOICE OF $p(K)$

To efficiently optimize a limit, one should choose an estimator to minimize the product of the second moment of the *gradient* estimates and the expected compute cost per evaluation. The choice of $p(K)$ effects both the variance and computation cost of our estimator. Denoting $\hat{G} := \nabla_\theta \hat{Y}$ and $\Delta_k^g := \nabla_\theta[\text{IWAE}_{k+1}(x) - \text{IWAE}_k(x)]$, the Russian roulette estimator is optimal across a broad family of unbiased randomized truncation estimators if the $\Delta_k^g$ are statistically independent, in which case it has second moment $\mathbb{E}||\hat{G}||_2^2 = \sum_{k=1}^{\infty} \mathbb{E}||\Delta_k^g||_2^2/\mathbb{P}(\mathcal{K} \geq k)$ (Beatson & Adams, 2019). While the $\Delta_k^g$ are not in fact strictly independent with our sampling procedure (Algorithm 1), and other estimators within the family may perform better, we justify our choice by showing that $\mathbb{E}\Delta_i\Delta_j$ for $i \neq j$ converges to zero much faster than $\mathbb{E}\Delta_k^2$ (Appendices A.2 & A.3). In the following, we assume independence of $\Delta_k^g$ and choose $p(K)$ to minimize the product of compute and variance.

---

**Algorithm 1** Computing SUMO, an unbiased estimator of $\log p(x)$.

**Input:** $x$, $m \geq 1$, encoder $q(z; x)$, decoder $p(x, z)$, $p(K)$, `reverse_cdf`$(\cdot) = \mathbb{P}(\mathcal{K} \geq \cdot)$
1: Sample $K \sim p(\mathcal{K})$
2: Sample $\{z_k\}_{k=1}^{K+m} \overset{iid}{\sim} q(z; x)$
3: $\log w_k \leftarrow \log p(x, z_k) - \log q(z_k; x)$
4: `ks` $\leftarrow [1, \ldots, K + m]$
5: `cum_iwae` $\leftarrow$ `log_cumsum_exp`$(\log w_k) - \log(\text{ks}[:k+1])$
6: `inv_weights` $= 1/\text{reverse\_cdf}(\text{ks})$
**return** `cum_iwae[m-1]` + `sum(inv_weights * (cum_iwae[m:] - cum_iwae[m-1:-1]))`

---

We first show that $\mathbb{E}||\Delta_k^g||_2^2$ is $\mathcal{O}(1/k^2)$ (Appendix A.5). This implies the optimal compute-variance product (Rhee & Glynn, 2015; Beatson & Adams, 2019) is given by $\mathbb{P}(\mathcal{K} \geq k) \propto \sqrt{\mathbb{E}||\Delta_k^g||_2^2}$. In our case, this gives $\mathbb{P}(\mathcal{K} \geq k) = 1/k$, which results in an estimator with infinite expected computation and no finite bound on variance. In fact, any $p(K)$ which gives rise to provably finite variance requires a heavier tail than $\mathbb{P}(\mathcal{K} \geq k) = 1/k$ and so will have infinite expected computation.

Though we could not theoretically show that our estimator and gradients have finite variance, we empirically find that gradient descent converges —even in the setting of minimizing log probability. We plot $||\Delta_k||_2^2$ for the toy variational inference task used to assess signal to noise ratio in Tucker et al. (2018) and Rainforth et al. (2018b), and find that they converge faster than $\frac{1}{k^2}$ in practice (Appendix A.6). While this indicates the variance is better than the theoretical bound, an estimator having infinite expected computation cost will always be an issue as it indicates significant probability of sampling arbitrarily large $K$. We therefore modify the tail of the sampling distribution such that the estimator has finite expected computation:

$$\mathbb{P}(\mathcal{K} \geq k) = \begin{cases} 1/k & \text{if } k < \alpha \\ 1/\alpha \cdot (1 - 0.1)^{k-\alpha} & \text{if } k \geq \alpha \end{cases} \qquad (10)$$

We typically choose $\alpha = 80$, which gives an expected computation cost of approximately 5 terms.

### 3.2.1 TRADING VARIANCE AND COMPUTE

One way to improve the RR estimator is to construct it so that some minimum number of terms (denoted here as $m$) are always computed. This puts a lower bound on the computational cost, but can potentially lower variance, providing a design space for trading off estimator quality against computational cost. This corresponds to a choice of RR estimator in which $\mathbb{P}(\mathcal{K} = K) = 0$ for $K \leq m$. This computes the sum out to $m$ terms (effectively computing $\text{IWAE}_m$) and then estimates the remaining difference with Russian roulette:

$$\text{SUMO}(x) = \text{IWAE}_m(x) + \sum_{k=m}^{K} \frac{\Delta_k(x)}{\mathbb{P}(\mathcal{K} \geq k)}, \quad K \sim p(K) \qquad (11)$$

In practice, instead of tuning parameters of $p(K)$, we set $m$ to achieve a given expected computation cost per estimator evaluation for fair comparison with IWAE and related estimators.

### 3.3 TRAINING $q(z; x)$ TO REDUCE VARIANCE

The SUMO estimator does not require amortized variational inference, but the use of an "encoder" to produce an approximate posterior $q(z; x)$ has been shown to be a highly effective way to perform rapid feedforward inference in neural latent variable models. We use $\phi$ to denote the parameters of the encoder $q_\phi(z; x)$. However, the gradients of SUMO with respect to $\phi$ are in expectation zero precisely because SUMO is an unbiased estimator of $\log p_\theta(x)$, regardless of our choice of $q_\phi(z; x)$. Nevertheless, we would expect the choice of $q_\phi(z; x)$ significantly impacts the variance of our estimator. As such, we optimize $q_\phi(z; x)$ to reduce the variance of the SUMO estimator. We can obtain unbiased gradients in the following way (Ruiz et al., 2016; Tucker et al., 2017):

$$\nabla_\phi \text{Var}[\text{SUMO}] = \nabla_\phi \mathbb{E}[\text{SUMO}^2] - \nabla_\phi(\mathbb{E}[\text{SUMO}])^2 = \mathbb{E}[\nabla_\phi \text{SUMO}^2]. \qquad (12)$$

Notably, the expectation of this estimator depends on the variance of SUMO, which we have not been able to bound. In practice, we observe gradients which are sometimes very large. We apply gradient clipping to the encoder to clip gradients which are excessively large in magnitude. This helps stabilize the training progress but introduces bias into the encoder gradients. Fortunately, the encoder itself is merely a tool for variance reduction, and biased gradients with respect to the encoder can still significantly help optimization.

### 3.4 APPLICATIONS OF UNBIASED LOG PROBABILITY

Here we list some applications where an unbiased log probability is useful. Using SUMO to replace existing lower bound estimates allows latent variable models to be used for new applications where a lower bound is inappropriate. As latent variable models can be both expressive and efficient to sample from, they are frequently useful in applications where the data is high-dimensional and samples from the model are needed.

**Minimizing** $\log p_\theta(x)$. Some machine learning objectives include terms that seek to increase the entropy of the learned model. The "reverse KL" objective—often used for training models to perform approximate posterior inference—minimizes $\mathbb{E}_{x \sim p_\theta(x)}[\log p_\theta(x) - \log \pi(x)]$ where $\pi(x)$ is a target density that may only be known up a normalization constant. Local updates of this form are the basis of the expectation propagation procedure (Minka, 2001). This objective has also been used for distilling autoregressive models that are inefficient at sampling (Oord et al., 2018). Moreover, reverse KL is connected to the use of entropy-regularized objectives (Williams & Peng, 1991; Ziebart, 2010; Mnih et al., 2016; Norouzi et al., 2016) in decision-making problems, where the goal is to encourage the decision maker toward exploration and prevent it from settling into a local minimum.

**Unbiased score function** $\nabla_\theta \log p_\theta(x)$. The score function is the gradient of the log-likelihood with respect to the parameters and has uses in estimating the Fisher information matrix and performing stochastic gradient Langevin dynamics (Welling & Teh, 2011), among other applications. Of particular note, the REINFORCE gradient estimator (Williams, 1992)—generally applicable for optimizing objectives of the form $\max_\theta \mathbb{E}_{x \sim p_\theta(x)}[R(x)]$—is estimated using the score function. This can be replaced with the gradient of SUMO which itself is an estimator of the score function $\nabla_\theta \log p_\theta(x)$.

$$
\begin{aligned}
\nabla_\theta \mathbb{E}_{x \sim p_\theta(x)}[R(x)] &= \mathbb{E}_{x \sim p_\theta(x)}[R(x) \nabla_\theta \log p_\theta(x)] \\
&= \mathbb{E}_{x \sim p_\theta(x)}[R(x) \nabla_\theta \mathbb{E}[\text{SUMO}(x)]] \\
&= \mathbb{E}_{x \sim p_\theta(x)}[\mathbb{E}[R(x) \nabla_\theta \text{SUMO}(x)]]
\end{aligned}
\tag{13}
$$

where the inner expectation is over the stochasticity of the SUMO estimator. Such estimators are often used for reward maximization in reinforcement learning where $p_\theta(x)$ is a stochastic policy.

## 4 RELATED WORK

There is a long history in Bayesian statistics of marginal likelihood estimation in the service of model selection. The harmonic mean estimator (Newton & Raftery, 1994), for example, has a long (and notorious) history as a consistent estimator of the marginal likelihood that may have infinite variance (Murray & Salakhutdinov, 2009) and exhibits simulation psuedo-bias (Lenk, 2009). The Chib estimator (Chib, 1995), the Laplace approximation, and nested sampling (Skilling, 2006) are alternative proposals that can often have better properties (Murray & Salakhutdinov, 2009). Annealed importance sampling (Neal, 2001) probably represents the gold standard for marginal likelihood estimation. These, however, turn into consistent estimators at best when estimating the *log* marginal probability (Rainforth et al., 2018a). Bias removal schemes such as jackknife variational inference (Nowozin, 2018) have been proposed to debias log-evidence estimation, IWAE in particular. Hierarchical IWAE (Huang et al., 2019) uses a joint proposal to induce negative correlation among samples and connects the convergence of variance of the estimator and the convergence of the lower bound.

Russian roulette also has a long history. It dates back to unpublished work from von Neumann and Ulam, who used it to debias Monte Carlo methods for matrix inversion (Forsythe & Leibler, 1950) and particle transport problems (Kahn, 1955). It has gained popularity in statistical physics (Spanier & Gelbard, 1969; Kuti, 1982; Wagner, 1987), for unbiased ray tracing in graphics and rendering (Arvo & Kirk, 1990), and for a number of estimation problems in the statistics community (Wei & Murray, 2017; Lyne et al., 2015; Rychlik, 1990; 1995; Jacob & Thiery, 2015; Jacob et al., 2017). It has also been independently rediscovered many times (Fearnhead et al., 2008; McLeish, 2011; Rhee & Glynn, 2012; Tallec & Ollivier, 2017).

The use of Russian roulette estimation in deep learning and generative modeling applications has been gaining traction in recent years. It has been used to solve short-term bias in optimization problems (Tallec & Ollivier, 2017; Beatson & Adams, 2019). Wei & Murray (2017) estimates the reciprocal normalization constant of an unnormalized density. Han et al. (2018) uses a similar random truncation approach to estimate the distribution of eigenvalues in a symmetric matrix. Along similar motivations with our work, Chen et al. (2019) uses this estimator to construct an unbiased estimator of the change of variables equation in the context of normalizing flows (Rezende & Mohamed, 2015), and Xu et al. (2019) uses it to construct unbiased log probability for a nonparameteric distribution in the context of variational autoencoders (Kingma & Welling, 2014).

Though we extend latent variable models to applications that require unbiased estimates of log probability and benefit from efficient sampling, an interesting family of models already fulfill these requirements. Normalizing flows (Rezende & Mohamed, 2015; Dinh et al., 2017) offer exact log probability and certain models have been proven to be universal density estimators (e.g. Huang et al. (2018)). However, these models often require restrictive architectural choices with no dimensionality-reduction capabilities, and make use of many more parameters to scale up (Kingma & Dhariwal, 2018) than alternative generative models. Discrete variable versions of these models are still in their infancy and make use of biased gradients (Tran et al., 2019; Hoogeboom et al., 2019), whereas latent variable models naturally extend to discrete observations.

## 5 DENSITY MODELING EXPERIMENTS

We first compare the performance of SUMO when used as a replacement to IWAE with the same expected cost on density modeling tasks. We make use of two benchmark datasets: dynamically binarized MNIST (LeCun et al., 1998) and binarized OMNIGLOT (Lake et al., 2015).

We use the same neural network architecture as IWAE (Burda et al., 2016). The prior $p(z)$ is a 50-dimensional standard Gaussian distribution. The conditional distributions $p(x_i|z)$ are independent Bernoulli, with the decoder parameterized by two hidden layers, each with 200 $\tanh$ units. The approximate posterior $q(z; x)$ is also a 50-dimensional Gaussian distribution with diagonal covariance, whose mean and variance are both parameterized by two hidden layers with 200 $\tanh$ units. We reimplemented and tuned IWAE, obtaining strong baseline results which are better than those previously reported. We then used the same hyperparameters to train with the SUMO estimator. We find clipping very large gradients can help performance, as large gradients may be infrequently sampled. This introduces a small amount of bias into the gradients while reducing variance, but can nevertheless help achieve faster convergence and should still result in a less-biased estimator. A posthoc study of the effect on final test performance as a function of this bias-variance tradeoff mechanism is discussed in Appendix A.7. We note that gradient clipping is only done for the density modeling experiments.

The averaged test log-likelihoods and standard deviations over 3 runs are summarized in Table 1. To be consistent with existing literature, we evaluate our model using IWAE with 5000 samples. In all the cases, SUMO achieves slightly better performance than IWAE with the same expected cost. We also bold the results that are statistically insignificant from the best performing model according to an unpaired $t$-test with significance level 0.05. However, we do see diminishing returns as we increase $k$, suggesting that as we increase compute, the variance of our estimator may impact performance more than the bias of IWAE.

## 6 LATENT VARIABLES MODELS FOR ENTROPY MAXIMIZATION

We move on to our first task for which a lower bound estimate of log probability would not suffice. The reverse KL objective is useful when we have access to a (possibly unnormalized) target

Table 1: Test negative log-likelihood of the trained model, estimated using IWAE($k$=5000). For SUMO, $k$ refers to the expected number of computed terms.

| Training Objective | MNIST | | | OMNIGLOT | | |
|---|---|---|---|---|---|---|
| | $k$=5 | $k$=15 | $k$=50 | $k$=5 | $k$=15 | $k$=50 |
| ELBO (Burda et al., 2016) | 86.47 | — | 86.35 | 107.62 | — | 107.80 |
| IWAE (Burda et al., 2016) | 85.54 | — | 84.78 | 106.12 | — | 104.67 |
| ELBO (Our impl.) | 85.97±0.01 | 85.99±0.05 | 85.88±0.07 | 106.79±0.08 | 106.98±0.19 | 106.84±0.13 |
| IWAE (Our impl.) | 85.28±0.01 | 84.89±0.03 | 84.50±0.02 | 104.96±0.04 | 104.53±0.05 | **103.99±0.12** |
| JVI (Our impl.) | — | — | 84.75±0.03 | — | — | 104.08±0.11 |
| SUMO | **85.09±0.01** | **84.71±0.02** | **84.40±0.03** | **104.85±0.04** | **104.29±0.12** | **103.79±0.14** |

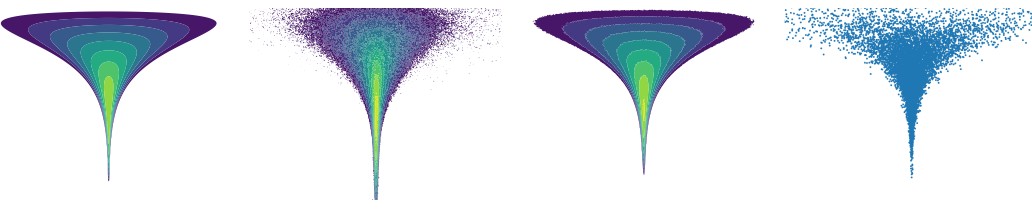

Target log probability     Training w/ IWAE (k=15)   Training w/ SUMO (k=15)      Model samples

Figure 1: We trained latent variable models for posterior inference, which requires minimizing log probability under the model. Training with IWAE leads to optimizing for the bias while leaving the true model in an unstable state, whereas training with SUMO—though noisy—leads to convergence.

distribution but no efficient sampling algorithm.

$$\min_{\theta} D_{\mathrm{KL}}(p_{\theta}(x)||p^*(x)) = \min_{\theta} \mathbb{E}_{x \sim p_{\theta}(x)}[\log p_{\theta}(x) - \log p^*(x)] \tag{14}$$

A major problem with fitting latent variables models to this objective is the presence of an entropy maximization term, effectively a minimization of $\log p_{\theta}(x)$. Estimating this log marginal probability with a lower bound estimator could result in optimizing $\theta$ to maximize the bias of the estimator instead of the true objective. Our experiments demonstrate that this causes IWAE to often fail to optimize the objective unless we use a large amount of computation.

**Modifying IWAE.** The bias of the IWAE estimator can be interpreted as the KL between an importance-weighted approximate posterior $q_{IW}(z;x)$ implicitly defined by the encoder and the true posterior $p(z|x)$ (Domke & Sheldon, 2018). Both the encoder and decoder parameters can therefore affect this bias. In practice, we find that the encoder optimization proceeds at a faster timescale than the decoder optimization: i.e., the encoder can match $q_{IW}(z;x)$ to the decoder's posterior $p(z|x)$ more quickly than the latter can match an objective. For this reason, we train the encoder to reduce bias and use a minimax training objective

$$\min_{p(x,z)} \max_{q(z;x)} \mathbb{E}_{x \sim p(x)}[\mathrm{IWAE}_K(x) - \log p^*(x)] \tag{15}$$

Though this is still a lower bound with unbounded bias, it makes for a stronger baseline than optimizing $q(z;x)$ in the same direction as $p(x,z)$. We find that this approach can indeed work in practice, but requires setting $k$ extremely high.

We choose a "funnel" target distribution (Figure 1) similar to the distribution used as a benchmark for inference in Neal et al. (2003), where $p^*$ has support in $\mathbb{R}^2$ and is defined $p^*(x_1, x_2) = \mathcal{N}(x_1; 0, 1.35^2)\mathcal{N}(x_2; 0, e^{2x_1})$ We use neural networks with one hidden layer of 200 hidden units and $\tanh$ activations for both the encoder and decoder networks. We use 20 latent variables, with $p(z)$, $p_{\theta}(x|z)$, and $q_{\phi}(z;x)$ all being Gaussian distributed.

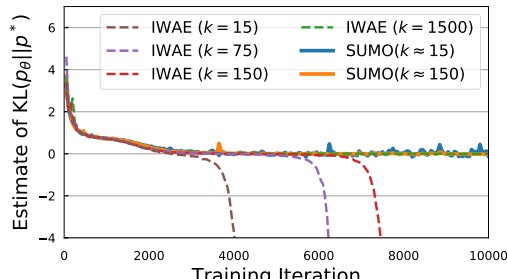

Figure 2: Training with reverse KL requires minimizing $\log p(x)$. SUMO estimates are unbiased and trains well, but minimizing the lower bound IWAE with small $k$ leads to estimates of $-\infty$.

Figure 2 shows the learning curves when using IWAE and SUMO. Unless $k$ is set very large, IWAE will at some point start optimizing the bias instead of the actual objective. The reverse KL is a non-negative quantity, so any estimate significantly below zero can be attributed to the unbounded bias. On the other hand, SUMO correctly optimizes for the objective even with a small expected cost. Increasing the expected cost $k$ for SUMO simply reduces variance. We also found that if $k$ is set sufficiently large, then IWAE can work when we train using the minimax objective in equation 15, suggesting that a sufficiently debiased estimator can also work in practice. However, this requires much more compute and likely does not scale compared to SUMO. We also visualize the contours of the resulting models in Figure 1. For IWAE, we visualize the model a few iterations before it reaches numerical instability.

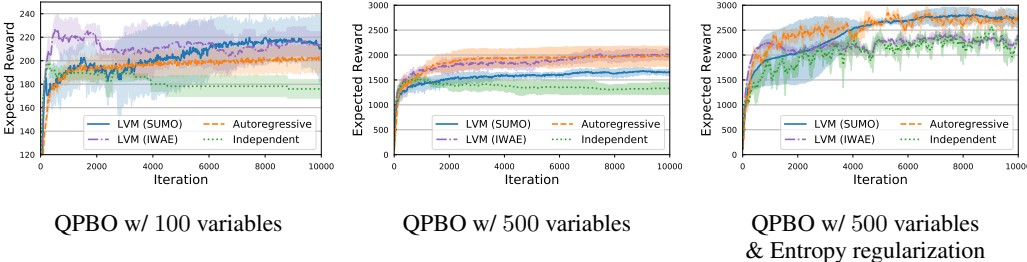

QPBO w/ 100 variables        QPBO w/ 500 variables        QPBO w/ 500 variables
& Entropy regularization

Figure 3: Latent variable policies allow faster exploration than autoregressive policy models, while being more expressive than an independent policy. SUMO works well with entropy regularization, whereas IWAE is unstable and converges to similar performance as the non-latent variable model.

## 7 LATENT VARIABLE POLICIES FOR COMBINATORIAL OPTIMIZATION

Let us now consider the problem of finding the maximum of a non-differentiable function, a special case of reinforcement learning without an interacting environment. Variational optimization (Staines & Barber, 2012) can be used to reformulate this as the optimization of a parametric distribution,

$$\max_x R(x) \geq \max_\theta \mathbb{E}_{x \sim p_\theta(x)}[R(x)], \tag{16}$$

which is now a differentiable function with respect to the parameters $\theta$, whose gradients can be estimated using a combination of the REINFORCE gradient estimator and the SUMO estimator (equation 13). Furthermore, entropy regularized reinforcement learning—where we maximize $R(x) + \lambda \mathcal{H}(p_\theta)$ with $\mathcal{H}(p_\theta)$ being the entropy of $p_\theta(x)$—encourages exploration and is inherently related to minimizing a reverse KL objective with the target being an exponentiated reward (Norouzi et al., 2016).

For concreteness, we focus on the problem of quadratic pseudo-Boolean optimization (QPBO) where the objective is to maximize

$$R(x) = \sum_{i=1} w_i(x_i) + \sum_{i<j} w_{ij}(x_i, x_j) \tag{17}$$

where $\{x_i\}_{i=1}^d \in \{0, 1\}$ are binary variables. Without further assumptions, QPBO is NP-hard (Boros & Hammer, 2002). As there exist complex dependencies between the binary variables and optimization of equation 16 requires sampling from the policy distribution $p_\theta(x)$, a model that is both expressive and allows efficient sampling would be ideal. For this reason, we motivate the use of latent variable models with independent conditional distributions, trained using the SUMO estimator. Our baselines are an autoregressive policy, which captures dependencies but for which sampling must be performed sequentially, and an independent policy, which is easy to sample from but captures no dependencies.

$$p_{\text{LVM}}(x) \coloneqq \int \prod_{i=1}^d p_\theta(x_i|z)p(z)dz \qquad p_{\text{Autoreg}}(x) \coloneqq \prod_{i=1}^d p(x_i|x_{<i}) \qquad p_{\text{Indep}}(x) \coloneqq \prod_{i=1}^d p(x_i)$$

We note that Haarnoja et al. (2018) also argued for latent variable policies in favor of learning diverse strategies but ultimately made use of normalizing flows which do not require marginalization.

We constructed one problem instance for each $d \in \{100, 500\}$, which we note are already intractable for exact optimization. For each instance, we randomly sampled the weights $w_i$ and $w_{ij}$ uniformly from the interval $[-1, 1]$. Figure 3 shows the performance of each policy model. In general, the independent policy is quick to converge to a local minima and is unable to explore different regions, whereas more complex models have a better grasp of the "frontier" of reward distributions during optimization. The autoregressive model works well overall but is much slower to train due to its sequential sampling procedure; with $d = 500$, it is $19.2\times$ slower per iteration than SUMO. Surprisingly, we find that estimating the REINFORCE gradient with IWAE results in decent performance when no entropy regularization is present. With entropy regularization, all policies improve

significantly; however, training with IWAE in this setting results in performance similar to the independent model, suggesting that it may not be using the latent state. On the other hand, SUMO works with both REINFORCE gradient estimation and entropy regularization, albeit at the cost of slower convergence due to variance.

# 8 CONCLUSION

We introduced SUMO, a new unbiased estimator of the log probability for latent variable models, and demonstrated tasks for which this estimator performs better than standard lower bounds. Specifically, we investigated applications involving entropy maximization where a lower bound performs poorly, but our unbiased estimator can train properly with relatively smaller amount of compute.

In the future, we plan to investigate new families of gradient-based optimizers which can handle heavy-tailed stochastic gradients. It may also be fruitful to investigate the use of convex combination of consistent estimators within the SUMO approach, as any convex combination is unbiased, or to apply variance reduction methods to increase stability of training with SUMO.

## ACKNOWLEDGEMENTS

This work was partially funded by NSF IIS-1421780. Y.L and J.Z were supported by the NSF China Project (No. 61620106010), Beijing NSF Project (No. L172037), the JP Morgan Faculty Research Program and the NVIDIA NVAIL Program with GPU/DGX Acceleration.

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

## A  APPENDIX

### A.1  DERIVATION OF SUMO

Let

$$\mathbb{E}_q[\text{IWAE}_k(x)] = \mathbb{E}_{z_1,\ldots,z_k \sim q(z;x)}\left[\log \frac{1}{K}\sum_{k=1}^{K}\frac{p_\theta(x \mid z_k)\,p_\theta(z_k)}{q(z_k;x)}\right],$$

where $z_1, .., z_k$ are sampled independently from $q(z;x)$. And we define the $k$-th term of the infinite series $\tilde{\Delta}_k(x) \coloneqq \mathbb{E}_q[\text{IWAE}_{k+1}(x)] - \mathbb{E}_q[\text{IWAE}_k(x)]$. Using the properties of IWAE in equation 6, we have $\tilde{\Delta}_k(x) \geq 0$, and

$$\sum_{k=1}^{\infty}|\tilde{\Delta}_k(x)| = \sum_{k=1}^{\infty}\tilde{\Delta}_k(x) = \lim_{k\to\infty}\mathbb{E}_q[\text{IWAE}_k(x)] - \mathbb{E}_q[\text{IWAE}_1(x)]$$
$$= \log p_\theta(x) - \mathbb{E}_q[\text{IWAE}_1(x)] < \infty, \tag{18}$$

which means the series converges absolutely. This is a sufficient condition for finite expectation of the Russian roulette estimator (Chen et al. (2019); Lemma 3). Applying equation 7 to the series:

$$\log p_\theta(x) = \mathbb{E}_q[\text{IWAE}_1(x)] + \sum_{k=1}^{\infty}\tilde{\Delta}_k(x) \tag{19}$$

$$= \mathbb{E}_q[\text{IWAE}_1(x)] + \mathbb{E}_{K\sim p(K)}\left[\sum_{k=1}^{K}\frac{\tilde{\Delta}_k(x)}{\mathbb{P}(\mathcal{K}\geq k)}\right] \tag{20}$$

$$= \mathbb{E}_{K\sim p(K)}\left[\mathbb{E}_q[\text{IWAE}_1(x)] + \sum_{k=1}^{K}\frac{\mathbb{E}_q[\text{IWAE}_{k+1}(x) - \text{IWAE}_k(x)]}{\mathbb{P}(\mathcal{K}\geq k)}\right] \tag{21}$$

$$= \mathbb{E}_{K\sim p(K)}\left[\mathbb{E}_q\left[\text{IWAE}_1(x) + \sum_{k=1}^{K}\frac{\text{IWAE}_{k+1}(x) - \text{IWAE}_k(x)}{\mathbb{P}(\mathcal{K}\geq k)}\right]\right]. \tag{22}$$

Let $\Delta_k(x) \coloneqq \text{IWAE}_{k+1}(x) - \text{IWAE}_k(x)$, Hence our estimator is constructed:

$$\text{SUMO}(x) = \text{IWAE}_1(x) + \sum_{k=1}^{K}\frac{\Delta_k(x)}{\mathbb{P}(\mathcal{K}\geq k)}, \quad K\sim p(K), z_k \overset{iid}{\sim} q(z;x)\,. \tag{23}$$

And it can be easily seen from equation 22 and equation 23 that SUMO is an unbiased estimator of the log marginal likelihood:

$$\mathbb{E}_{K\sim p(K),z_1,\ldots,z_K\sim q(z;x)}\left[\text{SUMO}(x)\right] = \log p_\theta(x). \tag{24}$$

### A.2  CONVERGENCE OF $\Delta_k$

We follow the analysis of JVI (Nowozin, 2018), which applied the delta method for moments to show the asymptotic results on the bias and variance of $\text{IWAE}_k$ both at a rate of $\mathcal{O}(\frac{1}{k})$. We build on this analysis to analyze the convergence of $\Delta_k$.

Let $w_i = \frac{p(x|z_i)p(z_i)}{q(z_i;x)}$ and we define $Y_k \coloneqq \frac{1}{k}\sum_{i=1}^{k}w_i$ as the sample mean and we have $\mathbb{E}[Y_k] = \mathbb{E}[w] = \mu$.

$$\text{IWAE}_k = \log Y_k = \log\left[\mu + (Y_k - \mu)\right]$$
$$= \log\mu - \sum_{t=1}^{\infty}\frac{(-1)^t}{t\mu^t}(Y_k - \mu)^t \tag{25}$$

We note that we rely on $||Y_k - \mu|| < 1$ for this power series to converge. This condition was implicitly assumed, but not explicitly noted, in (Nowozin, 2018). This condition will hold for sufficiently

large $k$ so long as the moments of $w_i$ exist: one could bound the probability $||Y_k - \mu|| \geq 1$ by Chebyshev's inequality or by the Central Limit Theorem. We use the central moments $\gamma_t := \mathbb{E}[(Y_k - \mu)^t]$ and $\mu_t := \mathbb{E}[(w - \mu)^t]$ for $t \geq 2$.

$$\mathbb{E}\Delta_k^2 = \mathbb{E}(\text{IWAE}_{k+1} - \text{IWAE}_k)^2 \tag{26}$$

$$= \mathbb{E}\left[\log\mu - \sum_{t=1}^{\infty}\frac{(-1)^t}{t\mu^t}(Y_{k+1} - \mu)^t - \log\mu + \sum_{t=1}^{\infty}\frac{(-1)^t}{t\mu^t}(Y_k - \mu)^t\right]^2 \tag{27}$$

$$= \mathbb{E}\left[\sum_{t=1}^{\infty}\frac{(-1)^t}{t\mu^t}\left[(Y_k - \mu)^t - (Y_{k+1} - \mu)^t\right]\right]^2 \tag{28}$$

Expanding Eq. 28 to order two gives

$$\mathbb{E}\Delta_k^2 = \mathbb{E}\left[-\frac{1}{\mu}(Y_k - \mu - Y_{k+1} + \mu) + \frac{1}{2\mu^2}\left[(Y_k - \mu)^2 - (Y_{k+1} - u)^2\right]\right]^2 + o(k^{-2}) \tag{29}$$

$$= \frac{1}{\mu^2}\mathbb{E}\left[Y_{k+1} - Y_k + \frac{1}{2\mu}(Y_k + Y_{k+1} - 2\mu)(Y_k - Y_{k+1})\right]^2 + o(k^{-2}) \tag{30}$$

$$= \frac{1}{\mu^2}\mathbb{E}\left[2(Y_{k+1} - Y_k) + \frac{1}{2\mu}(Y_k + Y_{k+1})(Y_k - Y_{k+1})\right]^2 + o(k^{-2}) \tag{31}$$

Since we use cumulative sum to compute $Y_k$ and $Y_{k+1}$, we obtain $Y_{k+1} = \frac{kY_k + w_{k+1}}{k+1}$.

$$\implies \mathbb{E}\Delta_k^2 = \frac{1}{\mu^2}\mathbb{E}\left[2\frac{w_{k+1} - 1}{k+1} + \left(\frac{w_{k+1} + \frac{2k+1}{k+1}\sum_{i=1}^{k}w_k}{2k\mu}\right)\left(\frac{w_{k+1} - 1}{k+1}\right)\right]^2 + o(k^{-2}) \tag{32}$$

We note that $\frac{w_{k+1} - 1}{k+1} = \mathcal{O}(\frac{1}{k})$ and $\frac{w_{k+1} + \frac{2k+1}{k+1}\sum_{i=1}^{k}w_k}{2k\mu} = \mathcal{O}(1)$. Therefore $\Delta_k$ is $\mathcal{O}(\frac{1}{k})$, and $\mathbb{E}\Delta_k^2 = \mathcal{O}(\frac{1}{k^2})$.

## A.3 Convergence of $\Delta_k\Delta_j$

Without loss of generality, suppose $j \geq k + 1$,

$$\mathbb{E}\Delta_k\Delta_j = \mathbb{E}\left[\left(\sum_{t=1}^{\infty}\frac{(-1)^t}{t\mu^t}\left[(Y_k - \mu)^t - (Y_{k+1} - \mu)^t\right]\right)\left(\sum_{t=1}^{\infty}\frac{(-1)^t}{t\mu^t}\left[(Y_j - \mu)^t - (Y_{j+1} - \mu)^t\right]\right)\right] \tag{33}$$

For clarity, let $C_k = Y_k - \mu$ be the zero-mean random variable. Nowozin (2018) gives the relations

$$\mathbb{E}[C_k^2] = \gamma_2 = \frac{\mu_2}{k} \tag{34}$$

$$\mathbb{E}[C_k^3] = \gamma_3 = \frac{\mu_3}{k^2} \tag{35}$$

$$\mathbb{E}[C_k^4] = \gamma_4 = \frac{3\mu_2^2}{k^2} + \frac{\mu_4 - 3\mu_2^2}{k^3} \tag{36}$$

$$\mathbb{E}\Delta_k\Delta_j = \mathbb{E}\left[\left(\sum_{t=1}^{\infty}\frac{(-1)^t}{t\mu^t}(C_k^t - C_{k+1}^t)\right)\left(\sum_{t=1}^{\infty}\frac{(-1)^t}{t\mu^t}(C_j^t - C_{j+1}^t)\right)\right] \tag{37}$$

Expanding both the sums inside the brackets to order two:

$$\mathbb{E}\Delta_k\Delta_j \approx \mathbb{E}\frac{1}{\mu^2}(C_{k+1} - C_k)(C_{j+1} - C_j) \qquad (1)$$

$$- \mathbb{E}\frac{1}{2\mu^3}(C_{k+1}^2 - C_k^2)(C_{j+1} - C_j) \qquad (2)$$

$$- \mathbb{E}\frac{1}{2\mu^3}(C_{k+1} - C_k)(C_{j+1}^2 - C_j^2) \qquad (3)$$

$$+ \mathbb{E}\frac{1}{4\mu^4}(C_{k+1}^2 - C_k^2)(C_{j+1}^2 - C_j^2) \qquad (4)$$

We will proceed by bounding each of the terms $(1)$, $(2)$, $(3)$, $(4)$. First, we decompose $C_j$. Let $B_{k,j} := \frac{1}{j}\sum_{i=k+1}^{j}(w_i - \mu)$.

$$C_j = \frac{1}{j}\left(kC_k + \sum_{i=k+1}^{j}(w_i - \mu)\right) = \frac{k}{j}C_k + \frac{1}{j}\sum_{i=k+1}^{j}(w_i - \mu) = \frac{k}{j}C_k + B_{k,j} \qquad (38)$$

We know that $B_{k,j}$ is independent of $C_k$ and $\mathbb{E}[B_{k,j}] = 0$, implying $\mathbb{E}[C_k B_{k,j}] = 0$. Note $C_j^2 = \frac{k^2}{j^2}C_k^2 + 2\frac{k}{j}C_k B_{k,j} + B_{k,j}^2$.

Now we show that $(1)$ is zero:

$$\mathbb{E}[\frac{1}{\mu^2}(C_{k+1} - C_k)(C_{j+1} - C_j)] = \frac{1}{\mu^2}\mathbb{E}\Big[C_{k+1}\frac{k+1}{j+1} + C_{k+1}B_{j+1,k+1}$$
$$- \frac{k+1}{j}C_{k+1}^2 - B_{j,k+1}C_{k+1} - C_k\frac{k}{j+1}$$
$$- C_k B_{j+1,k} + \frac{k}{j}C_k^2 + C_k B_{j,k}\Big]$$
$$= \frac{1}{\mu^2}\mathbb{E}\Big[-\frac{k+1}{j(j+1)}C_{k+1}^2 + C_k^2\frac{k}{j(j+1)}\Big]$$
$$= \frac{1}{\mu^2}\Big[-\frac{k+1}{j(j+1)}\frac{\mu_2}{k+1} + \frac{\mu_2}{k}\frac{k}{j(j+1)}\Big] = 0$$

We now investigate $(2)$:

$$\mathbb{E}[-\frac{1}{2\mu^3}(C_{k+1}^2 - C_k^2)(C_{j+1} - C_j)] = \frac{1}{2\mu^3}\mathbb{E}\Big[C_k^3\frac{k}{j+1} + C_k^2 B_{j+1,k} - C_k^3\frac{k}{j} - C_k^2 B_{j,k}$$
$$+ C_{k+1}^3\frac{k+1}{j+1} + C_{k+1}^2 B_{j,k} + \frac{k}{j}C_k^2 + C_k B_{j+1,k}\Big]$$
$$= \frac{1}{\mu^2}\mathbb{E}[-\frac{k+1}{j(j+1)}C_{k+1}^2 + C_k^2\frac{k}{j(j+1)}]$$
$$= \frac{1}{2\mu^3}\Big[-\frac{\mu_3}{kj(j+1)} + \frac{\mu_3}{(k+1)j(j+1)}\Big] = -\frac{\mu_3}{2\mu^3}\Big[\frac{1}{k(k+1)j(j+1)}\Big]$$

We now show that $(3)$ is zero:

$$\mathbb{E}[\frac{1}{2\mu^3}(C_{k+1} - C_k)(C_j^2 - C_{j+1}^2)] = \frac{1}{2\mu^3}\mathbb{E}[C_{k+1}C_j^2 - C_{k+1}C_{j+1}^2 - C_k C_j^2 + C_k C_{j+1}^2)]$$
$$= \frac{1}{2\mu^3}\Big[\frac{\mu_3}{j^2} - \frac{\mu_3}{(j+1)^2} - \frac{\mu_3}{j^2} - \frac{\mu_3}{(j+1)^2}\Big]$$
$$= 0$$

Finally, we investigate $(4)$:

$$\mathbb{E}[\frac{1}{4\mu^4}(C_{k+1}^2 - C_k^2)(C_{j+1}^2 - C_j^2)]$$

Using the relation in equation 36, we have

$$\mathbb{E}[C_k^2 C_j^2] = \mathbb{E}[C_k^2(\frac{k^2}{j^2}C_k^2 + \frac{2k}{j}C_k B_{j,k} + B_{j,k}^2)] \qquad (39)$$
$$= \frac{k^2}{j^2}\gamma_4 + \gamma_2\frac{(j-k)\mu_2}{j^2} \qquad (40)$$
$$= \frac{(2k+j-3)\mu_2^2 + \mu_4}{j^2 k} \qquad (41)$$

$$\mathbb{E}[\frac{1}{4\mu^4}(C_{k+1}^2 - C_k^2)(C_{j+1}^2 - C_j^2)] = \frac{(2k+j-3)\mu_2^2 + \mu_4}{j^2k} - \frac{(2k+j-2)\mu_2^2 + \mu_4}{(j+1)^2k}$$

$$- \frac{(2k+j-1)\mu_2^2 + \mu_4}{j^2(k+1)} + \frac{(2k+j)\mu_2^2 + \mu_4}{(j+1)^2(k+1)}$$

$$= \frac{(j^2 - 5j - 3)\mu_2^2}{j^2(j+1)^2k(k+1)} - \frac{\mu_4}{(j+1)^2k(k+1)}$$

$$= \frac{j^2(\mu_2^2 - \mu_4) - (5j+3)\mu_2^2}{j^2(j+1)^2k(k+1)}$$

$$= \mathcal{O}(j^{-2}k^{-2})$$

In summary, $\mathbb{E}\Delta_k\Delta_j$ is $\mathcal{O}(k^{-2}j^{-2})$.

### A.4   GRADIENT OF SUMO

Assume that $\nabla_\theta \text{SUMO}$ is bounded: it is sufficient that $\nabla_\theta \text{IWAE}_1$ is bounded and that the sampling probabilities are chosen such that the partial sums of $\frac{\nabla_\theta \Delta_k}{\mathbb{P}(\mathcal{K} \geq k)}$ converge, i.e. $\mathbb{P}(\mathcal{K} \geq k) > ck||\nabla_\theta \Delta_k||$ for some constant $c$. Then we have $\mathbb{E}\left[\nabla_\theta \text{SUMO}(x)\right] = \nabla_\theta \mathbb{E}\left[\text{SUMO}(x)\right] = \nabla_\theta \log p_\theta(x)$ directly by the dominated convergence theorem, as long as SUMO is everywhere differentiable, which is satisfied by all of our experiments. If ReLU neural networks are to be used, one may be able to show the same property using Theorem 5 of Bikowski et al. (2018), assuming finite higher moments and Lipschitz continuity.

### A.5   CONVERGENCE OF $\nabla\left(\text{IWAE}_{k+1} - \text{IWAE}_k\right)$

The IWAE log likelihood estimate is:

$$\mathcal{L}_k = \log\left(\frac{1}{k}\sum_{i=1}^{k}\frac{p_\theta(x, z_i)}{q_\psi(z_i|x)}\right)$$

The gradient of this with respect to $\lambda$, where $\lambda$ is either $\theta$ or $\psi$, is

$$\frac{d\mathcal{L}_k}{d\lambda} = \frac{1}{\sum_{i=1}^{k}\frac{p_\theta(x,z_i)}{q_\psi(z_i|x)}}\sum_{i=1}^{k}\frac{d}{d\lambda}\frac{p_\theta(x, z_i)}{q_\psi(z_i|x)}$$

We abbreviate $w_i := \frac{p_\theta(x,z_i)}{q_\psi(z_i|x)}$, and $\nu_i = \frac{dw_i}{d\lambda}$. In both $\lambda = \psi$ and $\lambda = \theta$ cases, it suffices to treat the $w_i$ and $\nu_i$ as i.i.d. random variables with finite variance and expectation. Being a likelihood ratio, $w_i$ could be ill behaved when the importance sampling distribution $q_\psi(z_i|x)$ is is particularly mismatched from the true posterior $p(z_i|x) = \frac{p_\theta(x,z_i)}{\mathbb{E}_{z\sim p(z)}p_\theta(x,z)}$. However, the analysis from IWAE (Burda et al., 2016) requires assuming that the likelihood ratios $w_i = \frac{p_\theta(x,z_i)}{q_\psi(z_i|x)}$ are bounded, and we adopt this assumption. Reasoning about when this assumption holds, and the behavior of IWAE-like estimators when it does not, is an interesting area for future work.

Consider the differences between two gradients: we label $\Delta^g$ as follows:

$$\Delta_k^g := \frac{d\mathcal{L}_{k+1}}{d\lambda} - \frac{d\mathcal{L}_k}{d\lambda}$$

We have:

$$\Delta_k^g = \frac{1}{\sum_{i=1}^{k+1} w_i}\nu_{k+1} + \left(\frac{1}{\sum_{i=1}^{k+1} w_i} - \frac{1}{\sum_{i=1}^{k} w_i}\right)\sum_{i=1}^{k}\nu_i$$

$$= \frac{1}{\sum_{i=1}^{k+1} w_i}\nu_{k+1} + \frac{w_{k+1}}{\left(\sum_{i=1}^{k+1} w_i\right)\left(\sum_{i=1}^{k} w_i\right)}\sum_{i=1}^{k}\nu_i$$

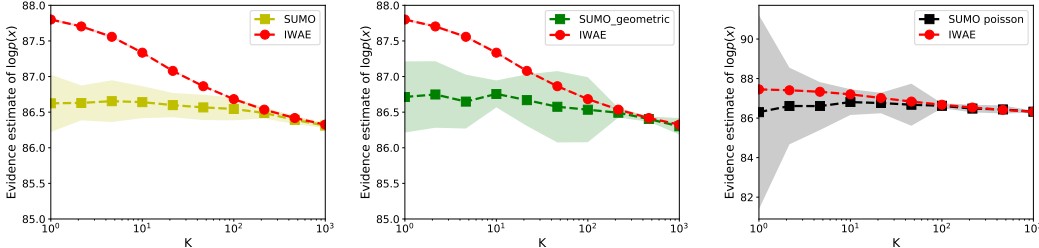

Figure 4: A comparison of SUMO estimations with different distributions and IWAE estimations of test negative log-likelihood on a trained model with IWAE$_1$ objective on MNIST. The expected cost is $K + 5$ for each evaluation. The results are averaged over 100 runs (mean in bold and std shaded).

We again let $Y_k$ denote the $k$th sample mean $\frac{1}{k}\sum_i w_i$. Then:

$$\Delta_k^g = \frac{1}{kY_k}\nu_{k+1} + \frac{w_{k+1}}{(k+1)Y_kY_{k+1}}\bar{\nu}_k$$

The sample means $Y_k$ and $\bar{\mu}_k$ have finite expectation and variance. The variance vanishes as $k \to \infty$ (but the expectation does not change).

$$\mathbb{E}||\Delta_k^g||_2^2 = \frac{1}{k^2}\mathbb{E}||\frac{\nu_{k+1}}{Y_k} + \frac{k}{k+1}\frac{w_{k+1}\bar{\nu}_k}{Y_kY_{k+1}}||_2^2$$

$$\text{Let } \frac{\nu_{k+1}}{Y_k} + \frac{k}{k+1}\frac{w_{k+1}\bar{\nu}_k}{Y_kY_{k+1}} := \phi_k$$

$$\implies \mathbb{E}||\Delta_k^g||_2^2 = \frac{1}{k^2}||\mathbb{E}\phi_k||_2^2 + \frac{1}{k^2}\text{Var}(\phi_k)$$

The second term vanishes at a rate strictly faster than $\frac{1}{k^2}$: the variance of $\phi_k$ goes to zero as $k \to \infty$. But the first term does not: $\phi_k$ is a biased estimator of $\phi_\infty$ so $\mathbb{E}\phi_k$ does change with $k$, but it does not necessarily go to zero:

$$\mathbb{E}\phi_\infty = \mathbb{E}\left[\frac{\nu}{\mathbb{E}w} + \frac{k}{k+1}\frac{w\mathbb{E}\nu}{(\mathbb{E}w)^2}\right] = \frac{\mathbb{E}\nu}{\mathbb{E}w}$$

Thus, $\mathbb{E}||\Delta_k^g||_2^2$ is at most $\mathcal{O}(\frac{1}{k^2})$.

## A.6 EMPIRICAL CONFIRMATION ON THE CONVERGENCE OF $\mathbb{E}\Delta_k^2$ AND $\mathbb{E}||\Delta_k^g||_2^2$

We measure the $\Delta_k^2$ and $||\Delta_k^g||_2^2$ on a toy example to verify the convergence rates empirically. We re-implement the toy Gaussian example from Rainforth et al. (2018b); Tucker et al. (2018). The generative model is $p_\theta(x, z) = \mathcal{N}(z|\theta, I)\mathcal{N}(x|z, I)$, where both $x$ and $z$ are in $\mathbb{R}^D$. The encoder is $q_\phi(z|x) = \mathcal{N}(z|Ax + b, \frac{2}{3}I)$, where $\phi = (A, b)$. The synthetic dataset was generated with $D = 20$ and $N = 1000$ data points using the true model parameter $\theta_{\text{true}}$ from a standard Gaussian. Alongside $||\Delta_k||_2^2$, we plot several reference convergence rates such as $\mathcal{O}(1/k^c)$, $c > 1$, and $\mathcal{O}(c^k)$, $c < 1$, as a visual guide. The results are shown in Figure 5. Following the setup in Rainforth et al. (2018b), we sample a group of model parameters close to the optimal values which are perturbed by Gaussian noise from $\mathcal{N}(0, 0.01^2)$. The gradient $\Delta_k^g$ is taken w.r.t. the model parameter $\theta$.

## A.7 BIAS-VARIANCE TRADEOFF VIA GRADIENT CLIPPING

While SUMO is unbiased, its variance is extremely high or potentially infinite. This property leads to poor performance compared to lower bound estimates such as IWAE when maximizing log-likelihood. In order to obtain models with competitive log-likelihood values, we can make use of gradient clipping. This allows us to ignore rare gradient samples with extremely large values due to the heavy-tailed nature of its distribution.

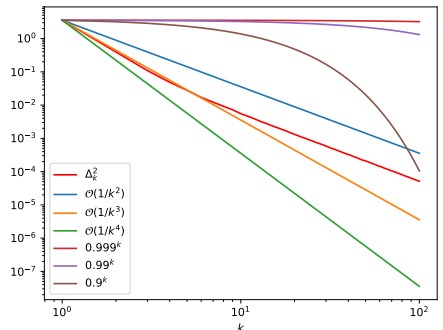 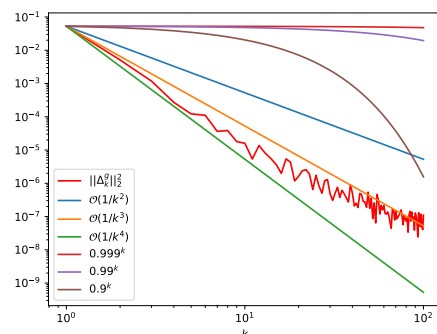

(a) Mean of estimated of $\mathbb{E}\Delta_k^2$ with increasing $k$ over ten random trials with 1000 samples per trial. X and Y axis are on log scale. Empirically the convergence rate of $\Delta_k^2$ is between $\mathcal{O}(1/k^2)$ and $\mathcal{O}(1/k^3)$.

(b) Mean of estimated $\mathbb{E}||\Delta_k^g||_2^2$ with increasing $k$ over ten trials with 1000 samples per trial. Empirically the convergence is faster than theoretical analysis $\mathcal{O}(1/k^2)$.

Figure 5: Empricial validation of the convergence rate of the norms of $\Delta$ and $\Delta^g$.

Gradient clipping introduces bias in favor of reduced variance. Figure 6 shows how the performance changes as a function of the clipping value, and more importantly, the percentage of clipped gradients. As shown, neither full clipping nor no clipping are desirable. We performed this experiment after reporting the results in Table 1, so this grid search was not used to tune hyperparameters for our experiments. As bias is introduced, we do not use gradient clipping for entropy maximization or policy gradient (REINFORCE).

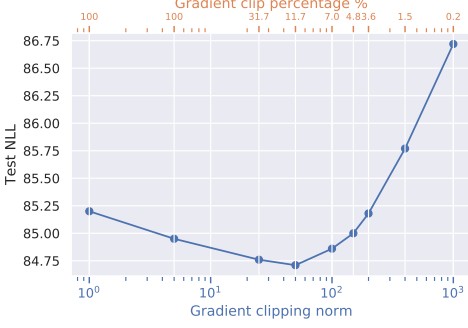 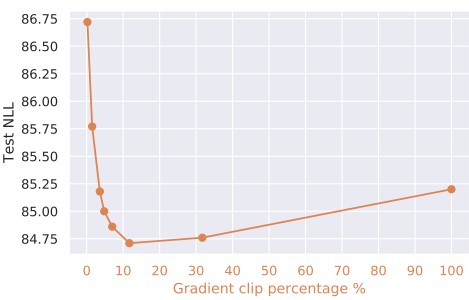

Figure 6: Test negative log-likelihood against the gradient clipping norm and clipping percentage, when training with SUMO (k=15).

## A.8 EXPERIMENTAL SETUP

In density modeling experiments, all the models are trained using a batch size of 100 and the AMS-Grad optimizer (Reddi et al., 2018) with parameters $lr = 0.001$, $\beta_1 = 0.9$, $\beta_2 = 0.999$ and $\epsilon = 10^{-4}$. The learning rate is reduced by factor $0.8$ if the validation likelihood does not improve for 50 epochs. We use gradient norm scaling in both the inference and generative networks. We train SUMO using the same architecture and hyperparameters as IWAE except the gradient clipping norm. We set the gradient norm to 5000 for encoder and $\{20, 40, 60\}$ for decoder in SUMO. For IWAE, the gradient norm is fixed to 10 in all the experiments. We report the performance of models with early stopping if no improvements have been observed for 300 epochs on the validation set.

We add additional plots of the test NLL against the norm and percentage of gradients clipped for the decoder in Figure 6. The plot is based on MNIST with expected number of compute $k = 15$. Gradient clipping was not used in the other experiments except the density modeling ones, where it was used as a simple tool to obtain a better bias-variance trade-off.

### A.8.1 REVERSE KL AND COMBINATORIAL OPTIMIZATION

These two tasks use the same encoder and decoder architecture: one hidden layer with $\tanh$ non-linearities and 200 hidden units. We set the latent state to be of size 20. The prior is a standard Gaussian with diagonal covariance, while the encoder distribution is a Gaussian with parameterized diagonal covariance. For reverse KL, we used independent Gaussian conditional likelihoods for $p(x|z)$, while for combinatorial optimization we used independent Bernoulli conditional distributions. We found it helps stablize training for both IWAE and SUMO to remove momentum and used RMSprop with learning rate 0.00005 and epsilon 1e-3 for fitting reverse KL. We used Adam with learning rate 0.001 and epsilon 1e-3, plus standard hyperparameters for the combinatorial optimization problems. SUMO used an expected compute of 15 terms.

