# OpenReview forum: "SUMO: Unbiased Estimation of Log Marginal Probability for Latent Variable Models"
_ICLR.cc/2020/Conference — Accept (Spotlight)_

### Official Review · AnonReviewer3 · 2019-10-17
**Official Blind Review #3**

**Rating:** 8

**Review:**

This paper presents an unbiased estimator of marginal log likelihood given a latent variable model.
The method extends the importance-weighted log marginal using the Russian roulette estimator.
The marginal log probability estimator is motivated for entropy maximized density estimation and use of REINFORCE (log-derivative) gradient for learning a policy with a latent variable.

The paper is well-organized and provides a contribution for optimizing latent variable models in certain scenarios.
Thus, I vote for its acceptance.

Some questions follow.
1) Is it trivial to show the absolute convergence of \Delta_k(x) series?
The absolute convergence is mentioned above equation (8), I am not convinced of this point.
Perhaps, if its expectation with respect to q(z;x) is applied, this can be shown from equation (6).
Otherwise we need some assumption on q(z;x) like q(z;x) is reasonably close to p(z) or p(z|x).

2) How was parameter m set for the experiments?

3) I assume the expectation operator is taken over z and K in equations (9, 12).
Is this correct? An explicit notation should be informative.

**Experience Assessment:**

I have read many papers in this area.

**Review Assessment: Checking Correctness Of Derivations And Theory:**

I assessed the sensibility of the derivations and theory.

**Review Assessment: Checking Correctness Of Experiments:**

I assessed the sensibility of the experiments.

**Review Assessment: Thoroughness In Paper Reading:**

I read the paper at least twice and used my best judgement in assessing the paper.

---

> ### Author Response · Authors · 2019-11-15
> **Response to Reviewer 3**
>
> Thank you for your careful review and valuable comments.
>
> 1) Yes, we apply this to the expected terms with expectation taken over q(z;x). We’ve added the derivation in Appendix A.1.
>
> 2) The expected computation cost is k=m + E[K] which was set to 5, 15, 50 for our experiments. For k=15, 50, we used the sampling distribution in Eq.(10), which has E[K] = 5, and we set m = k - 5. For the k=5 setting, we set K to be a geometric distribution with an expectation of 2 and set m=3.
>
> 3) Yes, you’re right. For now, we’ve placed a sentence to clarify this. A clearer notation is also provided in Appendix A.1.

---

### Official Review · AnonReviewer2 · 2019-10-22
**Official Blind Review #2**

**Rating:** 8

**Review:**

This paper proposes an unbiased estimator of $\log p_\theta(x)$. Many unbiased estimators of $p_\theta$ exist, but $\log p_\theta$ is needed in many other settings, some of which are not well-served by standard estimators of $p_\theta$. The SUMO estimator is essentially a Russian roulette-based extension of IWAE; it is exactly unbiased, but takes a random and unbounded number of samples.

This allows marginally better optimization of certain models than IWAE with a much smaller average number of samples, and (more importantly) opens new possibilities such as entropy maximization which are not well-served by lower bounds like IWAE. This is a very nice advantage of this estimator.

One complaint about this class of estimators in general is: yes, the exact SUMO procedure is technically unbiased. But in practice, if SUMO takes more than, say, a day of compute time – something that will happen with extremely small but nonzero probability – then the user will kill it. And the SUMO estimator conditioned on taking less than a day of compute time is actually biased. This also likely means that, though unbiased, these estimators can be potentially skewed or otherwise "unpleasant." For SUMO in particular, the estimator with $K$ truncated probably has bias bounded based on the bias of IWAE with batch size equal to the truncation point, which is likely quite small. But it would be nice to understand this a little more. (Perhaps it's been studied by some of the recent cited work on these types of estimators.)

Relatedly, you don't prove that this estimator has a finite variance, and in fact it seems plausible theoretically that it might be infinite. Like the "notorious" harmonic mean estimator, this is troubling. It seems that things are okay in practice, but how can we tell whether the variance is really finite or not? I don't know if there's a good answer, but one diagnostic might be something like the one suggested at https://stats.stackexchange.com/a/9143 . (Your comments about occasional "bounded but very large" gradient estimates are troubling in this respect, depending on what exactly you mean by "bounded".) When you do gradient clipping, the estimator of course then has a finite variance, but can we get some sense of how much bias that introduces?

Overall, though, I think this is a very nice new estimator that is both well-founded – despite leaving some questions open – and likely to be practically useful. Given that it is also extremely on-topic for ICLR and novel, I'm rating the paper as "accept."

(For slightly more detail on the "thoroughness assessment": I did not really check the proofs in the appendix, but did pay attention to the derivations in the main body.)


Smaller notes:

- Top of page 3: you comment that SGD "requires unbiased estimates of" gradients of the log-density. In fact, SGD can be shown to work with biased gradient estimators, with suboptimality in the results depending on the bias; see e.g. Chen and Luss, http://arxiv.org/abs/1807.11880 .

- In the definition of $\tilde{Y}$, above (7): it might make more sense to define $\tilde{Y}$ with some recursive scheme, rather than as an estimator that either computes one of the $\Delta$ values or infinitely many of them.

- Start of 3.1: presumably $\Delta_k$ is what converges absolutely, not IWAE?

- Start of 3.2: as you note, it is clearly not true that the $\Delta_k^g$ are independent. But you don't really "assume independence" – the Russian roulette estimator is still a valid estimator, just perhaps not the optimal among that class. It would be better to say something like that since it seems that the $\Delta_k^g$ are *nearly* independent (or at least nearly uncorrelated), the Russian roulette estimator is probably at least a reasonable choice.

- Re: the discussion after (9) and in (12), as well as a few other places: I think you show in Appendix A.3 that $\mathbb E[ \nabla \operatorname{SUMO}(x) ]$ exists, but you don't show that it equals $\nabla \mathbb E[ \operatorname{SUMO}(x) ]$. This is likely true, particularly if $q$ and $p$ are each everywhere-differentiable, and it's totally fine if you don't want to prove it out formally, but it would be worth at least a footnote that this is a thing that requires proof. (See e.g. https://arxiv.org/abs/1801.01401 Theorem 5, for a formal result of this type supporting ReLU activations, which you may be able to just use directly.)

**Experience Assessment:**

I have read many papers in this area.

**Review Assessment: Checking Correctness Of Derivations And Theory:**

I assessed the sensibility of the derivations and theory.

**Review Assessment: Checking Correctness Of Experiments:**

I assessed the sensibility of the experiments.

**Review Assessment: Thoroughness In Paper Reading:**

I read the paper at least twice and used my best judgement in assessing the paper.

---

> ### Author Response · Authors · 2019-11-15
> **Response to Reviewer 2**
>
> Thank you for going through the paper carefully and providing a positive feedback.
>
> Yes, there are some existing works on discussing the appearance of “pseudo-bias” due to never actually sampling a very large $K$. There is some discussion relating to the harmonic mean estimator, which we have now cited in the paper. While this is definitely a potential problem, we slightly argue that even if such bias is empirically present, it should be small enough to be ignored because SUMO is capable of minimizing log marginal probability in practice.
>
> Gradient clipping was not used in the other experiments except the density modeling ones, where we used it as a tool to obtain a better bias-variance tradeoff. In comparison, JVI has lower bias than IWAE but its high variance results in worse models, as reported by both the original paper and reflected in our experiments. For experiments involving entropy maximization, we found that any clipping results in unstable models. To better understand the bias-variance tradeoff, we have added a plot of the percentage of gradients clipped against the test NLL in Appendix A.6 (edit: A.7). We did not use this for grid-search as it was constructed after the submission. For our existing reported results, the percentage of clipped gradients is no more than 10%.
>
> Thank you for your small notes. We will update the paper accordingly. Below are some select responses:
>
> -$\Delta^g_k$ are *nearly* independent (or at least nearly uncorrelated)
> Yes, the discussion on independence is only because existing discussions on guaranteed finite variance and compute usually involve this assumption (e.g. [1,2]). We show in Appendices A.2 and A.3 that $\mathbb{E}[\Delta_i \Delta_j]$ for $i\neq j$ converges to zero much faster than $\mathbb{E}[\Delta_k^2]$.
>
> -$\nabla \mathbb{E}[\text{SUMO}(x)]$ and $\mathbb{E}[\nabla\text{SUMO}(x)]$:
> Yes, thank you for pointing this out. One can directly use the Dominated Convergence Theorem to show that $\nabla \mathbb{E}[\text{SUMO}(x)] = \mathbb{E}[\nabla\text{SUMO}(x)]$ so long as the function $\text{SUMO}(x)$ is differentiable with finite derivative for all $x$ at the current model parameters. The proof will be less direct when non-everywhere-differentiable activations such as ReLUs are used. However, theorem 5 from the paper you reference can be directly used to prove this works for ReLUs in our setting, given the mild assumptions used for that theorem (which are necessary for other estimators such as IWAE to work). We have added a discussion on this in the paper.
>
> [1] “Unbiased Estimation with Square Root Convergence for SDE Models”. Rhee and Glynn.
> [2] “Efficient optimization of loops and limits with randomized telescoping sums” Beatson and Adams.

---

> > ### Comment · AnonReviewer2 · 2019-11-15
> > **Thanks**
> >
> > Thanks for your replies; they all sound good. It looks like you haven't updated the pdf yet, though.
> >
> > The importance of gradient clipping in the different situations should be mentioned in the paper, as it seems to be practically important. I don't know exactly what's in your new Appendix A.6 but it sounds like that's a good step.

---

> > > ### Author Response · Authors · 2019-11-15
> > > **PDF now updated**
> > >
> > > Apologies, we did some last-minute editing after submitting the response. It is Appendix A.7 now. Thanks again for your valuable feedback.

---

### Official Review · AnonReviewer1 · 2019-10-25
**Official Blind Review #1**

**Rating:** 6

**Review:**

The authors consider the unbiased estimation of log marginal likelihood (evidence) after integration of latent variable. On top of the importance-weighted autoencoder (IWAE), which is only guaranteed to be asymptotically unbiased, the authors propose to use Russian Roulette estimator (RRE) to compensate the bias caused by the finite summation.

The proposed method is interesting and can be applied in many other estimators with similar properties as Eq. (6). Bias compensation using RRE is interesting, but it seems there must be many literatures that took advantage of using RRE to improve estimators. The authors have to be thorough in presenting previous research and explaining the authors’ contribution that is distinguished from those.

The authors showed synthetic and real application of the estimator, but one concern is the variance. Unbiasedness with finite samples often fails because of the variance, and regularization is often useful rather than correcting bias---If unbiasedness in important, regularization definitely breaks the unbiasedness---. The only discussion about variance is a few lines in page 3 after Eq. (7), but it is unclear how the variance problem is mitigated or why the problem does not suffer high variance.

In general, this paper is well-written and dealing with important problem with interesting method. Several analysis for understanding the advantages of using the proposed method is insufficient.


**Experience Assessment:**

I have published in this field for several years.

**Review Assessment: Checking Correctness Of Derivations And Theory:**

I assessed the sensibility of the derivations and theory.

**Review Assessment: Checking Correctness Of Experiments:**

I assessed the sensibility of the experiments.

**Review Assessment: Thoroughness In Paper Reading:**

I read the paper at least twice and used my best judgement in assessing the paper.

---

> ### Author Response · Authors · 2019-11-15
> **Response to Reviewer 1**
>
> Thank you for your thoughtful review and comments. We’ve added comparisons to previous work on bias reduced estimators (e.g. jackknife variational inference (JVI) (Nowozin, 2018)) in the related work and experiments. We have cited multiple bias compensation works with RRE, across multiple areas of application, as well as some key works on marginal likelihood estimation. As the main problem we’re tackling is the optimization of objectives involving the _log_ marginal likelihood, we felt our existing related work section correctly positions our contributions within the literature; however, we are open to suggestions.
>
> The regularization used during optimization (ie. gradient clipping) was only used for the density estimation tasks. We found this gave SUMO a good bias-variance tradeoff in terms of test performance, compared to existing biased estimators. On the other hand, JVI has theoretically lower bias than IWAE but a higher variance, and results in worse performance than IWAE. Experiments with SUMO on posterior inference and combinatorial optimization all used completely unbiased (gradient) estimators, because the bias in these situations can result in a “run-away” model which simply optimizes for the bias instead of the true objective. This effect happens for IWAE but not for SUMO (e.g. Figure 2).
>
> It was shown in Beatson & Adams, 2019 that bounded variance and compute is guaranteed if $|\Delta_k|^2$ vanishes faster than $O(k^{-2})$ and the sampling distribution is properly chosen. Empirically, we found that the variance is better than this theoretical bound required for finite variance. This empirical analysis is discussed in Appendix A.6. For now, we leave the theoretical proof of finite variance as an open problem and will look into it more in the future, as a thorough analysis can lead to improving the designs of SUMO or similar estimators.

---

### Decision · Program_Chairs · 2019-12-19

**Decision:**

Accept (Spotlight)

**Comment:**

The paper proposes a new way to train latent variable models. The standard way of training using the ELBO produces biased estimates for many quantities of interest. The authors introduce an unbiased estimate for the log marginal probability and its derivative to address this. The new estimator is based on the importance weighted autoencoder, correcting the remaining bias using russian roulette sampling. The model is empirically shown to give better test set likelihood, and can be used in tasks where unbiased estimates are needed.

All reviewers are positive about the paper. Support for the main claims is provided through empirical and theoretical results. The reviewers had some minor comments, especially about the theory, which the authors have addressed with additional clarification, which was appreciated by the reviewers.

The paper was deemed to be well organized. There were some unclarities about variance issues and bias from gradient clipping, which have been addressed by the authors in additional explanation as well as an additional plot.

The approach is novel and addresses a very relevant problem for the ICLR community: optimizing latent variable models, especially in situations where unbiased estimates are required. The method results in marginally better optimization compared to IWAE with much smaller average number of samples. The method was deemed by the reviewers to open up new possibilities such as entropy minimization.